# A Systematic Review and Meta-Analysis of Efficacy and Safety of Azithromycin Versus Moxifloxacin for the Initial Treatment of *Mycoplasma genitalium* Infection

**DOI:** 10.3390/antibiotics11030353

**Published:** 2022-03-07

**Authors:** Hideo Kato, Mao Hagihara, Nobuhiro Asai, Jun Hirai, Yuka Yamagishi, Takuya Iwamoto, Hiroshige Mikamo

**Affiliations:** 1Department of Clinical Infectious Diseases, Aichi Medical University, Nagakute 480-1195, Japan; katou.hideo.233@mail.aichi-med-u.ac.jp (H.K.); hagimao@aichi-med-u.ac.jp (M.H.); nobuhiro0204@gmail.com (N.A.); hiraichimed@gmail.com (J.H.); y.yamagishi@mac.com (Y.Y.); 2Department of Pharmacy, Mie University Hospital, Tsu 514-8507, Japan; taku-iwa@med.mie-u.ac.jp; 3Department of Clinical Pharmaceutics, Division of Clinical Medical Science, Mie University Graduate School of Medicine, Tsu 514-8507, Japan; 4Department of Molecular Epidemiology and Biomedical Sciences, Aichi Medical University Hospital, Nagakute 480-1195, Japan

**Keywords:** meta-analysis, *Mycoplasma genitalium*, azithromycin, moxifloxacin, initial treatment

## Abstract

*Mycoplasma genitalium* is recognized as a remarkable pathogen since azithromycin-resistant strains and treatment failure have been increasingly reported. Nevertheless, international guidelines still recommend azithromycin as a first-line treatment and moxifloxacin as a second-line treatment. We performed a systematic review and meta-analysis to validate the efficacy and safety of both drugs in the initial treatment of *M. genitalium*. We systematically searched the EMBASE, PubMed, Scopus, Ichushi, and CINAHL databases up to December 2021. We defined efficacy as clinical and microbiologic cure, and safety as persistent diarrhea. Overall, four studies met the inclusion criteria: one showed clinical cure (azithromycin treatment, n = 32; moxifloxacin treatment, n = 6), four showed microbiologic cure (n = 516; n = 99), and one showed safety (n = 63; n = 84). Moxifloxacin improved the microbiologic cure rate compared with azithromycin (odds ratio [OR] 2.79, 95% confidence interval [CI], 1.06–7.35). Clinical cure and safety did not show a significant difference between azithromycin and moxifloxacin treatments (OR 4.51, 95% CI 0.23–88.3; OR 0.63, 95% CI 0.21–1.83). Our meta-analysis showed that moxifloxacin was more effective than azithromycin at eradicating *M. genitalium* infections and supports its preferential use as a first-line treatment.

## 1. Introduction

*Mycoplasma genitalium* is a small bacterium belonging to the Mycoplasmataceae family and is implicated in the etiology of nongonococcal urethritis in men and cervicitis in women [1,2,3]. Although the prevalence of *M. genitalium* in a general population-based study of young adults is low (1%) [4], its prevalence in sexual health clinics is substantially higher, ranging from 10% to 20% [5,6,7]. Since *M. genitalium* does not have peptidoglycan-containing cell walls, treatment options are limited to antibiotics that disrupt protein synthesis (macrolides such as azithromycin, and tetracyclines such as doxycycline) or DNA replication (quinolones such as moxifloxacin). Clinical trials reported superior efficacy of azithromycin with a failure rate of 16% compared with doxycycline [8]. Moreover, it has been reported that the efficacy of moxifloxacin in patients with azithromycin treatment failure was 100%, with high in vitro susceptibility [9]. Therefore, international guidelines recommend azithromycin as a first-line treatment and moxifloxacin as a second-line treatment [10,11]. However, resistance to both antibiotics and treatment failure have recently been reported.

The eradication of *M. genitalium* is hampered by increased antibiotic resistance. Besides intrinsic resistance to all β-lactams, macrolide resistance has been reported in over 50% of *M. genitalium* isolates from patients with urethritis and cervicitis in many countries [12,13,14]. Moreover, azithromycin treatment fails in at least 10% of susceptible isolates, leading to the selection of strains with macrolide resistance-associated mutations (MRMs) at positions 2058 or 2059 in the 23S ribosomal RNA gene [15,16]. Quinolones are used as alternative agents against treatment failure in patients treated with azithromycin. However, a meta-analysis showed a decrease in the cure rate for moxifloxacin from 100% in studies until 2010 to 89% in studies from 2010 onward [17].

The Centers for Disease Control and Prevention (CDC) have developed a watch list of bacteria that show antibiotic-resistant threats with the potential to spread or become a challenge in the United States [18] and *M. genitalium* is included as a public health issue on that list. However, antibiotic resistance associated with treatment failure with azithromycin and moxifloxacin has not been reflected in the treatment strategy. We performed a systematic review and meta-analysis to validate the efficacy and safety of azithromycin and moxifloxacin as initial treatments for *M. genitalium* infections.

## 2. Results

### 2.1. Systematic Review

Data extracted from the electronic databases retrieved 865 potentially relevant articles. After removing duplicates, the titles and abstracts of 795 articles were screened. A full-text review of 19 articles was performed. Figure 1 depicts a full list of reasons for exclusion. Consequently, four studies met the inclusion criteria [9,19,20,21].

The characteristics of these studies are summarized in Table 1. One was a case-control study and the others were cohort studies. All studies were conducted in a single center; two in Australia [9,21], one in Norway [19] and one in Japan [20]. All participants reported by Bradshaw [9] and Terada [20] were patients with urethritis and cervicitis, respectively. The participants reported by Jernberg [19] and Gundevia [21] were patients with either urethritis or cervicitis. Although three of the four studies were conducted among patients aged 15–57 years old [9,20,21], the other study did not report the age of the patients included [19]. The susceptibility of the isolated *M. genitalium* was not reported in the four studies included in this review. A total of 516 patients were treated with azithromycin, while 99 were treated with moxifloxacin. The dosage regimens of azithromycin were as follows: 1 g single dose; 1 g weekly for 3 doses; 1 g single dose day 1, repeated after 5–7 days; 500 mg single dose day 1, 250 mg single dose for the following 4 days, 2 g single dose, 1 g single dose day 1, 500 mg single dose for the following 4 days. The moxifloxacin dosage regimen was 400 mg every 24 h, and the treatment duration was 7–14 days. Assessment of the risk-of-bias is shown in Table 1. The Newcastle–Ottawa Quality Assessment Scale score was 6.

### 2.2. Meta-Analysis

#### 2.2.1. Clinical Cure

One study reported clinical cure data of 32 patients treated with azithromycin and six patients treated with moxifloxacin [9]; the clinical cure rates in these two groups were 75% (24/32) and 100% (6/6), respectively. Moxifloxacin treatment did not improve the clinical cure rate compared with azithromycin treatment (OR 4.51, 95% CI 0.23–88.3).

#### 2.2.2. Microbiologic Cure

Microbiologic cure data from 516 patients treated with azithromycin and 99 patients treated with moxifloxacin were reported in four studies [9,19,20,21]. The microbiologic cure rates in the two groups were 77.5% (400/516) and 94.9% (94/99), respectively. Moxifloxacin treatment significantly improved the microbiologic cure rate when compared to the azithromycin treatment (OR 2.79, 95% CI 1.06–7.35, *I*^2^ = 0%; Figure 2).

#### 2.2.3. Safety

One study reported adverse events in 63 and 84 patients treated with azithromycin and moxifloxacin, respectively [20]. The incidence of adverse events was 12.7% (8/63) in patients treated with azithromycin and 8.3% (7/84) in those treated with moxifloxacin. The incidence of adverse events was not significantly different between the two antibiotic treatments (OR 0.63, 95% CI 0.21–1.83).

## 3. Discussion

A previous systematic review regarding the antibiotic susceptibility of *M. genitalium* and treatment efficacy of existing antibiotics has mentioned that moxifloxacin remains the most effective treatment despite the emergence of treatment failures and quinolone resistance [22]. However, most studies included in the review reported the efficacy of moxifloxacin treatment in patients who experienced azithromycin treatment failure. Therefore, we performed a meta-analysis to investigate the efficacy of azithromycin versus moxifloxacin as a first-line treatment for *M. genitalium* infection. Our meta-analysis showed superior microbiological cure rate in patients treated with moxifloxacin compared with patients treated with azithromycin. Moreover, all patients treated with moxifloxacin improved clinical cure, whereas 15% of patients treated with azithromycin did not improve clinical cure. Consequently, our findings indicated that moxifloxacin was a more effective first-line treatment for eradicating *M. genitalium* than azithromycin.

The eradication rate of *M. genitalium* from a single dose of 1 g azithromycin seems to decrease over time [23]. To date, there have been discussions about the most suitable dosing regimen of azithromycin for microbiologic cure in patients with *M. genitalium* infection. Previous studies reported that the eradication rate of *M. genitalium* in patients administered various dosing regimens of azithromycin showed no statistically significant difference between azithromycin 1 g in a single dose and other dosing regimens [19,20,21]. Hence, various azithromycin dosing regimens have been prescribed for the treatment of *M. genitalium* infections. Recently, a meta-analysis that reported the prevalence of mutations associated with resistance to macrolides in *M. genitalium* reported that the prevalence was significantly greater in the Americas than in the European region [24]. A conceivable cause is that the recommended dosing regimen based on treatment guidelines for *M. genitalium* is not standardized or optimized [3,25]. Therefore, global measures to optimize the efficacy of antibiotic treatments are urgently needed to prevent the further spread of macrolide-resistant strains.

A recent meta-analysis revealed that the prevalence of azithromycin-resistant *M. genitalium* increased from 10% before 2010 to 51% in 2016–2017, while that of moxifloxacin-resistant *M. genitalium* with *parC* (quinolone resistance-associated mutation, QRM) was 8% and did not change over time [24]. Among patients diagnosed with *M. genitalium* infection in 2017–2018, 64.4% had 23S ribosomal ribonucleic acid (rRNA) loci (MRMs), 11.5% had *parC*, and 0% had *gyrA* (QRM) [26]. The minimum inhibitory concentration (MIC) of azithromycin against all *M. genitalium* isolates with MRMs was over 8 mg/L [27,28]. According to these data, MRMs could contribute to increasing the MIC of azithromycin. Therefore, careful consideration of its use as a first-line treatment for *M. genitalium* infections is warranted. On the other hand, MICs of moxifloxacin against *M. genitalium* isolates with either *parC, gyrA,* or a mixture of both were 0.03–0.5 mg/L but adding MRMs to *M. genitalium* isolates with both QRMs led to moxifloxacin-resistant strains with MICs of over 2 mg/L [28]. Therefore, a single mutation in QRMs might not influence an increase in the MIC of moxifloxacin in *M. genitalium*. However, the correlation between *parC* and/or *gyrA* and moxifloxacin resistance is unclear because of limited data from cultured *M. genitalium*.

To date, two meta-analyses reported microbiological cure rates for infections due to *M. genitalium* [17,29]. According to the meta-analyses, the pooled microbiological cure rate of azithromycin was 67%, whereas that of moxifloxacin was 96%. In subgroup analyses of moxifloxacin, the pooled microbial cure rates for the initial use of moxifloxacin and the use of moxifloxacin after antibiotic treatment failure were 99% and 94%, respectively [29]. In the light of our findings, moxifloxacin can be effective as a first-line treatment as well as a second-line treatment for *M. genitalium* infections.

To our knowledge, this is the first meta-analysis to compare the efficacy and safety of azithromycin and moxifloxacin in patients with *M. genitalium* infections. However, our meta-analysis had several limitations. First, the number of studies included in our meta-analysis was relatively low and lacked information regarding patient backgrounds; therefore, subgroup analyses on various types of confounding factors, such as type of infection, susceptibility (mutation), and dosing regimen were not performed. Moreover, our meta-analysis only included single-center retrospective studies which might have increased the likelihood of reporting and selection bias. However, there was no heterogeneity (*I*^2^ = 0%) in the present study. Second, whether azithromycin and moxifloxacin were appropriate for the patients in the included studies was unclear because the results of susceptibility testing with the detected isolates were not reported. Indeed, culturing *M. genitalium* is difficult and is only performed in a few facilities worldwide. Finally, moxifloxacin cannot be prescribed for patients with *M. genitalium* infection in Japan because this antibiotic is only allowed to treat respiratory infections and skin and soft tissue infections. Instead, sitafloxacin, which is of the same generation as quinolone as moxifloxacin, is one of the standard treatments for *M. genitalium* infection in Japan. However, few studies have compared patients treated with azithromycin with those treated with sitafloxacin as a first-line treatment for *M. genitalium*.

## 4. Materials and Methods

### 4.1. Data Sources and Search Strategy

This study was performed in accordance with the PRISMA guidelines (Appendix A), except for the protocol registration on reporting systematic reviews and meta-analyses [30,31]. The following PICO criteria were used to select relevant studies: population (P), patients with *M. genitalium* infection; intervention (I), patients treated with azithromycin; comparison (C), patients treated with moxifloxacin; and outcome (O), efficacy and safety. All studies were identified through a systematic review of the EMBASE, PubMed, Scopus, Ichushi, and CINAHL databases until 1 December 2021, using the following terms: “*Mycoplasma genitalium*, macrolide, quinolone, urethritis,” and cervicitis.” Language restrictions were applied, except for English and Japanese.

### 4.2. Study Selection

Two authors independently reviewed articles based on titles and abstracts and then assessed the full-text articles. The full texts of eligible articles were reviewed to apply the inclusion criteria and identify articles for the final qualitative synthesis and meta-analysis. Any disagreement was resolved through discussion or based on the judgement of the third author (HM).

### 4.3. Eligibility Criteria

Studies that met the following criteria were extracted: (i) randomized controlled trials (RCTs), retrospective observational or cohort studies; (ii) patients diagnosed with urethritis or cervicitis infected with *M. genitalium*; and (iii) patients receiving azithromycin or moxifloxacin treatment. Studies reporting the efficacy of moxifloxacin for azithromycin treatment failure were excluded. No restrictions were placed on the antibiotic regimen or duration of antibiotic treatment. We evaluated the clinical cure, microbiologic cure, and safety of patients treated with azithromycin or moxifloxacin. A clinical cure was defined as the absence of signs and/or symptoms related to *M. genitalium* infection. A microbiologic cure was defined as a case when there was an absence of *M. genitalium* DNA or RNA after polymerase chain reaction (PCR) or transcription-mediated amplification of a urine sample, urethral or cervical swab, or biopsy specimen. The safety outcome was defined as the presence or absence of persistent diarrhea. 

### 4.4. Data Extraction and Risk-of-Bias Assessment

The following data were extracted: study design, setting, period, country of study, drug regimen, number of participants, age, type of infection, antibiotic susceptibility, and clinical outcome. The risk-of-bias was assessed using the Newcastle–Ottawa Quality Assessment tool for retrospective studies [32]. This scale consists of nine items assessing different study characteristics such as selection, comparability, and exposure. Two authors independently assessed the risk-of-bias.

### 4.5. Statistical Analysis

Our meta-analysis using Review Manager (RevMan, version 5.4; Nordic Cochrane StataCorp LLC, College Station, TX, USA) was performed according to a previous study [33]. Statistical heterogeneity between studies was evaluated using the chi-squared test. A *p*-value < 0.1 indicated significant heterogeneity. *I*^2^ represents the degree of heterogeneity (0–25%, low heterogeneity; 25–50%, moderate heterogeneity; 50–75%, substantial heterogeneity; and 75–100%, considerable heterogeneity). Heterogeneity was regarded as significant when *p* < 0.1, or *I*^2^ > 50%. A random-effects model was applied if the data were heterogeneous. In the other cases, a fixed-effects model was applied. The risk of clinical outcomes was calculated using odds ratios (ORs) and 95% confidence intervals (Cis) were calculated. The pooled Ors and 95% Cis were calculated using the fixed-effects and random-effects models, respectively, and the Ors from the results were compared.

## 5. Conclusions

In conclusion, our meta-analysis showed that moxifloxacin improved the microbiologic cure rate. The findings of this meta-analysis have provided evidence that azithromycin may be less effective than moxifloxacin in treating *M. genitalium* infection, whereas current guidelines recommend its use as a first-line treatment. However, national and international surveillance of antibiotic resistance in *M. genitalium* is needed to prevent the spread of moxifloxacin-resistant *M. genitalium* by inappropriate use of moxifloxacin. In the future, new treatment strategies, such as the development of novel antibiotics and antibiotic combination therapies, should be considered.

## Figures and Tables

**Figure 1 antibiotics-11-00353-f001:**
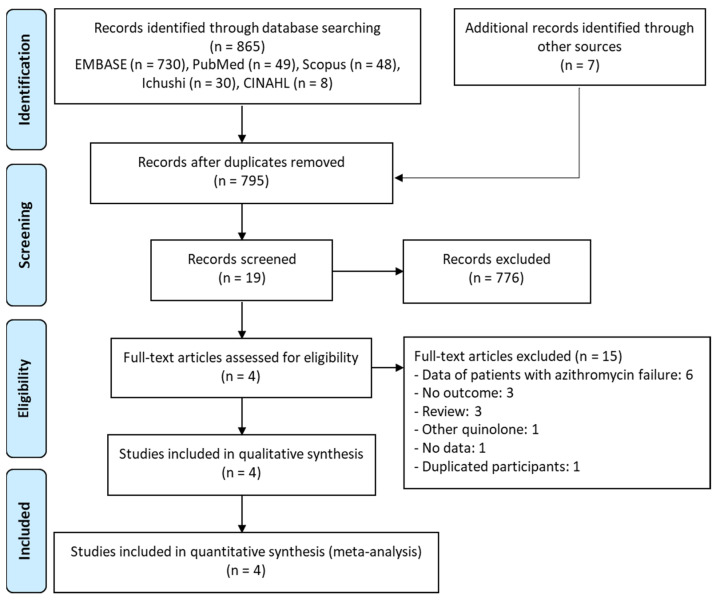
PRISMA flow diagram for the selection of eligible studies.

**Figure 2 antibiotics-11-00353-f002:**
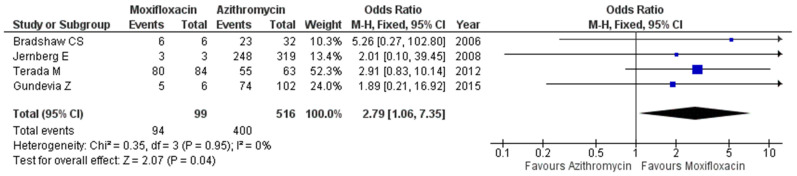
Forest plot presenting odds ratios for microbiologic cure comparing azithromycin and moxifloxacin in patients with *M. genitalium* infection.

**Table 1 antibiotics-11-00353-t001:** Characteristics of the studies included in the meta-analysis.

Study	Study Design	Setting	Period	Country of Study	Drug Regimen		No. of Participants	Age (Year)	Type of Infection	Susceptibility	Clinical Outcome	Risk-of-Bias Score
					Azithromycin	Moxifloxacin						
Bradshaw CS, 2006	Case-control	Single-center	Mar 2004 to Nov 2005	Australia	1 g single dose; 1 g weekly for 3 doses	400 mg every 24 h for 10 days	32 vs. 6	Median 33 (range 22–54)	Urethritis	NR	Microbiologic cure; Clinical cure	6
Jernberg E, 2008	Cohort	Single-center	May 2005 to Dec 2006	Norway	1 g single dose; 1 g single dose day 1, repeated after 5-7 days; 500 mg single dose day 1, 250 mg single dose the following 4 days	400 mg every 24 h for 7 days	319 vs. 3	NR	Urethritis; cervicitis	NR	Microbiologic cure	6
Terada M, 2012	Retrospective cohort	Single-center	Jan 2008 to Aug 2010	Japan	2 g single dose; 1 g single dose	400 mg every 24 h for 7 days; 400 mg every 24 h for 14 days	63 vs. 84	Range 18–42	Cervicitis	NR	Microbiologic cure; adverse event	6
Gundevia Z, 2015	Retrospective cohort	Single-center	Aug 2009 to May 2013	Australia	1 g single dose; 1 g single dose day 1, 500 mg single dose the following 4 days	NR	102 vs. 6	Mean 30 (range 15–57)	Urethritis; cervicitis	NR	Microbiologic cure	6

No, number; NR, not reported.

## Data Availability

All data are applicable in the paper.

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
