# Peer review of "A Systematic Review and Meta-Analysis of Efficacy and Safety of Azithromycin Versus Moxifloxacin for the Initial Treatment of Mycoplasma genitalium Infection"

_antibiotics, 2022, doi:10.3390/antibiotics11030353_

Round 1

Reviewer 1 Report

Thank you for the opportunity to review this interesting paper.

I have an issue withe the discussion - large part of the discussion goes back to the resistance profiles. This is not the core element in the paper, I think those paragraphs should be shortened.  In fact, no findings of the meta analysis are discussed.

Author list limitations, which I support, but fail to comunicate in the discussion on what does this meta analysis bring to the current knowledge on this important topic.

Author Response

  1. I have an issue with the discussion - large part of the discussion goes back to the resistance profiles. This is not the core element in the paper, I think those paragraphs should be shortened. In fact, no findings of the meta-analysis are discussed.

 Response: Thank you for your comment. We discussed as below: our meta-analysis showed superior microbiological cure rate in patients treated with moxifloxacin compared with patients with azithromycin. Moreover, all patients treated with moxifloxacin improved clinical cure, whereas 15% of patients treated with azithromycin did not improve clinical cure. Therefore, our findings indicated that moxifloxacin was a more effective first-line treatment for eradicating M. genitalium than azithromycin (p4 L122-126).

  1. Author list limitations, which I support, but fail to communicate in the discussion on what does this meta-analysis bring to the current knowledge on this important topic.

Response: Thank you for your comment. We discussed the picture regarding the use of moxifloxacin using two meta-analyses which reported microbiological cure rates for M. genitalium infections. In conclusion, we mentioned that moxifloxacin can be effective as a first-line treatment as well as a second-line treatment for M. genitalium infections (p5 L157-163).

Reviewer 2 Report

The manuscript "A systematic review and meta-analysis of efficacy and safety of azithromycin versus moxifloxacin for the initial treatment of Mycoplasma genitalium infection" by H. Kato et al. is an analysis of antibiotics treatment efficacy prepared according the PRISMA guidelines.

The work is written good and illustrated, it could be accepted for publication as is.

Author Response

The manuscript "A systematic review and meta-analysis of efficacy and safety of azithromycin versus moxifloxacin for the initial treatment of Mycoplasma genitalium infection" by H. Kato et al. is an analysis of antibiotics treatment efficacy prepared according the PRISMA guidelines.

The work is written good and illustrated, it could be accepted for publication as is.

Response: I appreciate for your comment.

Reviewer 3 Report

In this work, Kato et al., have done literature review and meta-analysis approach to compare the effectiveness of azithromycin and moxifloxacin as the first-line of treatment against Mycoplasma genitalium infection. The study is well conceived and the methodologies including the inclusion criteria are explained. While their conclusion is supported by the study, it is appreciated that the authors acknowledge the limitations of the work regarding their sample size, possible selection bias and the lack of information regarding the antibiotic susceptibility of the isolated strains from the patients. The article will be helpful to readers interested in M. genitalium treatment. There are a few minor suggestions which should benefit the manuscript.

  1. In the introduction, it will be helpful to discuss the mode of action of azithromycin and moxifloxacin.
  2. Bradshaw et al, 2006 (reference 20) study had 32 patients completing TOC in the azithromycin group. In the manuscript, Kato et al., mistakenly indicate in Table 1 under “No. of participants” that there were 35 vs 6 participants in the azithromycin vs moxifloxacin group. 35 should be corrected to 32 participants, i.e., 32 vs 6.
  3. Reference 17: PRISMA should be appropriately cited in the reference. It will be helpful if the authors cite one of the original publications instead of the website.
  4. Page 4 of 7, line 136-137 – Quinolone-resistance associated mutation (QRM) should be indicated in line 136.
  5. In the introduction, the authors briefly referred to Li et al., 2017 (reference 15) meta-analysis study that showed the gradual decrease of moxifloxacin effectiveness in the treatment of M. genitalium. In the light of the current finding, it would be helpful to discuss this previous study and provide the readers with a more complete picture regarding the use of moxifloxacin.

Author Response

Thank you for your reviewing our article. We made some changes in our manuscript according to reviewer’s suggestions with yellow highlight. We think some revises enhanced the quality of our manuscript. We had a mistake on numbering of reference from 17 and corrected the numbers.

In this work, Kato et al., have done literature review and meta-analysis approach to compare the effectiveness of azithromycin and moxifloxacin as the first-line of treatment against Mycoplasma genitalium infection. The study is well conceived and the methodologies including the inclusion criteria are explained. While their conclusion is supported by the study, it is appreciated that the authors acknowledge the limitations of the work regarding their sample size, possible selection bias and the lack of information regarding the antibiotic susceptibility of the isolated strains from the patients. The article will be helpful to readers interested in M. genitalium treatment. There are a few minor suggestions which should benefit the manuscript.

 Response: I appreciate for your comment.

  1. In the introduction, it will be helpful to discuss the mode of action of azithromycin and moxifloxacin.

 Response: Thank you for your comment. We added a short sentence regarding the mode of action of both antibiotics in Introduction (p1 L40-42).

  1. Bradshaw et al, 2006 (reference 20) study had 32 patients completing TOC in the azithromycin group. In the manuscript, Kato et al., mistakenly indicate in Table 1 under “No. of participants” that there were 35 vs 6 participants in the azithromycin vs moxifloxacin group. 35 should be corrected to 32 participants, i.e., 32 vs 6.

 Response: Thank you for your comment. We corrected the number of patients (Abstract, Table 1, p3 L94).

  1. Reference 17: PRISMA should be appropriately cited in the reference. It will be helpful if the authors cite one of the original publications instead of the website.

 Response: Thank you for your comment. We added one original publication regarding PRISMA (Reference 31).

  1. Page 4 of 7, line 136-137 – Quinolone-resistance associated mutation (QRM) should be indicated in line 136.

 Response: Thank you for your comment. The recent meta-analysis [24] defined moxifloxacin-resistant M. gentialium as isolates with parC. Hence, we added the information (p4 L144).

  1. In the introduction, the authors briefly referred to Li et al., 2017 (reference 15) meta-analysis study that showed the gradual decrease of moxifloxacin effectiveness in the treatment of M. genitalium. In the light of the current finding, it would be helpful to discuss this previous study and provide the readers with a more complete picture regarding the use of moxifloxacin.

 Response: Thank you for your comment. We discussed the picture regarding the use of moxifloxacin using two meta-analyses which reported microbiological cure rates for M. genitalium infections. In conclusion, we mentioned that moxifloxacin can be effective as a first-line treatment as well as a second-line treatment for M. genitalium infections (p5 L157-163).

Reviewer 4 Report

The study by Kato et al. reviewed the published literature on Azithromycin and Moxifloxacin to treat Mycoplasma genitaliun infection. This reviewer can, to some degree, understand the motivation of this study. However, the results and conclusions made by the authors were only based on a very limited number of studies and sample size, which might be biased, and not truly reflect the fact of those two medicines. The authors should also refer to the initial guideline and clinical trial data (which should support why those two antibiotics are recommended). Another major point is the effectors, such as age, gender, dose, duration, medical history, and anamnesis. Those are complex backgrounds and should not be simply explained by those limited samples. One minor issue is whether this article should be a review paper?

Author Response

Thank you for your reviewing our article. We made some changes in our manuscript according to reviewer’s suggestions with yellow highlight. We think some revises enhanced the quality of our manuscript. We had a mistake on numbering of reference from 17 and corrected the numbers.

  1. The study by Kato et al. reviewed the published literature on Azithromycin and Moxifloxacin to treat Mycoplasma genitaliun infection. This reviewer can, to some degree, understand the motivation of this study. However, the results and conclusions made by the authors were only based on a very limited number of studies and sample size, which might be biased, and not truly reflect the fact of those two medicines.

 Response: Thank you for your comment. In this meta-analysis, we focused on the efficacy of the two antibiotics as a first-line treatment. However, since most studies reported the efficacy of moxifloxacin for azithromycin treatment failure, we were able to include only four studies. Especially, clinical cure and safety was reported in one study, respectively. As you mentioned, our meta-analysis had several biases because of a limited number of studies and sample size, and therefore, we stated the limitation. However, there was no heterogeneity in our meta-analysis (p5 L166-172). On the other hand, microbiologic cure data from 516 patients treated with azithromycin and 99 patients treated with moxifloxacin was reported in four studies. Therefore, we concluded that moxifloxacin improved the microbiologic cure rate (p6 L236-239).

  1. The authors should also refer to the initial guideline and clinical trial data (which should support why those two antibiotics are recommended).

Response: Thank you for your comment. We added clinical trial data which supported that the two antibiotics are recommended by the guidelines (p1 L42-p2 L45).

  1. Another major point is the effectors, such as age, gender, dose, duration, medical history, and anamnesis. Those are complex backgrounds and should not be simply explained by those limited samples. One minor issue is whether this article should be a review paper?

Response: Thank you for your comment. As you mentioned, we thought that subgroup analyses evaluating whether various factors were associated with clinical outcomes were important to validation the efficacy and safety of the two antibiotics. However, we were unable to perform the subgroup analyses due to the lack of information regarding patient backgrounds (p5 L166-169). Moreover, our study is the first meta-analysis to compare the efficacy and safety of azithromycin and moxifloxacin in patients with M. genitalium infections. Therefore, our findings were reported as a meta-analysis paper not a review paper.

Round 2

Reviewer 1 Report

The manuscript has improved after the revision

Reviewer 4 Report

This reviewer has no additional comments on this revised manuscript.